# Distinct Changes in Gut Microbiota Are Associated with Estradiol-Mediated Protection from Diet-Induced Obesity in Female Mice

**DOI:** 10.3390/metabo11080499

**Published:** 2021-07-30

**Authors:** Kalpana D. Acharya, Hye L. Noh, Madeline E. Graham, Sujin Suk, Randall H. Friedline, Cesiah C. Gomez, Abigail E. R. Parakoyi, Jun Chen, Jason K. Kim, Marc J. Tetel

**Affiliations:** 1Neuroscience Department, Wellesley College, Wellesley, MA 02481, USA; kacharya@wellesley.edu (K.D.A.); mgraham2@wellesley.edu (M.E.G.); cgomez3@wellesley.edu (C.C.G.); aparakoy@wellesley.edu (A.E.R.P.); 2Program in Molecular Medicine, Division of Endocrinology, Metabolism, and Diabetes, Department of Medicine, University of Massachusetts Medical School, Worcester, MA 01605, USA; HyeLim.Noh@crl.com (H.L.N.); scarlet108@gmail.com (S.S.); Randall.Friedline@umassmed.edu (R.H.F.); Jason.Kim@umassmed.edu (J.K.K.); 3Department of Health Sciences Research & Center for Individualized Medicine, Mayo Clinic, Rochester, MN 55905, USA; Chen.Jun2@mayo.edu

**Keywords:** diabetes, estrogens, gut permeability/integrity, insulin sensitivity, *Akkermansia*, gut microbiome

## Abstract

A decrease in ovarian estrogens in postmenopausal women increases the risk of weight gain, cardiovascular disease, type 2 diabetes, and chronic inflammation. While it is known that gut microbiota regulates energy homeostasis, it is unclear if gut microbiota is associated with estradiol regulation of metabolism. In this study, we tested if estradiol-mediated protection from high-fat diet (HFD)-induced obesity and metabolic changes are associated with longitudinal alterations in gut microbiota in female mice. Ovariectomized adult mice with vehicle or estradiol (E2) implants were fed chow for two weeks and HFD for four weeks. As reported previously, E2 increased energy expenditure, physical activity, insulin sensitivity, and whole-body glucose turnover. Interestingly, E2 decreased the tight junction protein occludin, suggesting E2 affects gut epithelial integrity. Moreover, E2 increased *Akkermansia* and decreased Erysipleotrichaceae and Streptococcaceae. Furthermore, *Coprobacillus* and *Lactococcus* were positively correlated, while *Akkermansia* was negatively correlated, with body weight and fat mass. These results suggest that changes in gut epithelial barrier and specific gut microbiota contribute to E2-mediated protection against diet-induced obesity and metabolic dysregulation. These findings provide support for the gut microbiota as a therapeutic target for treating estrogen-dependent metabolic disorders in women.

## 1. Introduction

More than 40% of the US population is obese (CDC, 2018), which is a leading cause of morbidity and mortality worldwide [1]. The latest example of the increasing impact of obesity on human health is the strong association of obesity with the number of hospitalized COVID-19 positive patients [2]. Obesity is more prevalent in women [3], in particular during menopause, and is positively associated with a steep decline in ovarian hormones. Increased fat weight gain in postmenopausal women elevates their risk of hyperglycemia, insulin resistance, hyperlipidemia, low-grade inflammation, osteoporosis, cognitive decline, breast cancer and colorectal cancer [4,5,6,7,8,9]. Estrogen replacement therapy decreases the postmenopausal adiposity and protects women from diabetes, coronary heart disease, and increases overall lifespan [8,10]. Ovariectomized rodents provide excellent models for studying estrogen-dependent effects on energy homeostasis. Ovariectomy causes diet-induced obesity, hyperglycemia and insulin resistance in rodents, which can be rescued by estradiol (E2) treatment [11,12,13,14,15]. In further support of a protective role for estrogens, mice lacking estrogen receptors or the estrogen synthesizing enzyme, aromatase, develop obesity [16,17,18].

Another key regulator of energy homeostasis is the gut microbiota, a community of bacteria, fungi, viruses, and archaea that reside on the gastrointestinal epithelium [19]. The gut microbiota influences host physiology through nutrient harvest, synthesis of vitamins, hormones, and neurotransmitters, and imprinting and strengthening of the immune system [20,21]. Gut microbes produce energy from fermentation of non-digestible carbohydrates, in particular, short-chain fatty acids, that are linked to improved insulin sensitivity and health in humans [22]. Notably, germ-free mice and rats have a profound reduction in energy harvest capacity, a compromised immune system, and abnormal intestinal features compared to conventionally raised animals, indicating an important function of gut microbiota on host health. [23,24].

Diet strongly modulates gut microbial composition and activity in humans and rodents. For example, high-fat diet (HFD) profoundly decreases microbial diversity [21,25,26,27,28,29]. HFD promotes the endotoxin-producing gram-negative communities, inducing abnormal immune responses and inflammation, characteristic pathologies of obesity and diabetes [30,31]. HFD also increases intestinal permeability, allowing microbiota-induced toxins into the circulation and alters expression of multiple genes in the intestinal epithelium of male and female mice [31,32,33]. Given that HFD increases the risk of developing obesity and metabolic syndromes in postmenopausal women [34], it is important to gain a better understanding of the functions of gut microbiota in metabolic health in females.

Recent evidence suggests there is cross-talk between estrogens and gut microbiota. Urinary estrogens in postmenopausal women positively correlate with gut microbiota taxa diversity [35]. In further support of estrogens’ influence on gut microbiota in women, phytoestrogens increase *Lactobacillus*, *Enterococcus* and *Bifidobacterium* [36,37]. In mice, ovariectomy alters gut microbial diversity, in particular, by shifting abundances of the two major bacterial phyla, Bacteroidetes and Firmicutes and by increasing *Bifidobacterium* [38,39,40]. In mice fed a high-sucrose, high-fat containing western diet, chronic E2 administration via drinking water decreased lipopolysaccharide-producing microbes, such as *Escherichia* and *Shigella*, and increased *Bifidobacterium* and *Akkermansia* [41]. In addition, we recently found that estrogens alter gut microbiota in leptin-deficient (*ob/ob*) obese female mice. E2 decreased gut microbial evenness in both lean and obese (*ob/ob*) mice and increased S24-7 abundance [42]. Taken together, these studies suggest that estrogens can influence gut microbiota in females.

While diet and gut microbes profoundly affect energy metabolism, it is not known if the estrogen-mediated protective effects in females are linked to changes in gut microbiota. In the present study, we tested the hypothesis that the E2-mediated protection against HFD-induced obesity and metabolic disorders is associated with changes in gut microbiota and intestinal morphology in female mice. This study investigated the effects of E2 and diet in female mice on their metabolic profiles, associated longitudinal changes in gut microbiota, and gut epithelial integrity. These findings enhance our understanding of how estrogens function in women’s metabolic health and help identify potential gut microbial modulators in estrogen-dependent protection from metabolic syndromes.

## 2. Results

### 2.1. Estradiol Attenuates Body Weight and Fat Mass Gain in Female Mice on HFD

Ten-week-old female C57BL/6J mice were bilaterally ovariectomized (OVX) and received implants containing 17β-estradiol (E2) or vehicle (Veh, *n* = 6/group) [11,42]. Metabolic and gut microbiota data were collected at different points through the study (Figure 1A). Analysis of longitudinal data, including both STND and HFD feeding, showed a main effect of E2 on body weight. During the two weeks on STND, Veh and E2 mice did not differ in body weight. After switching to HFD, Veh mice gained weight, whereas E2 mice were protected from the weight gain. Veh mice weighed more than E2 mice from D21 till the end of the study (Figure 1B), due to increased fat mass (Figure 1C). The effect of E2 on fat mass was profound during HFD, although no effect was seen during STND. Lean mass was not affected by E2 during either diet (Figure 1D).

### 2.2. Estradiol Reduces Food Intake and Energy Expenditure in Female Mice on STND

Analysis of food intake (calories), including both STND and HFD feeding, showed a main effect of E2 treatment during night. During STND, E2 reduced food intake during 24 h and day (Figure 2A). An interaction between E2 and diet was also observed on food intake during 24 h, day and night. E2 altered water intake during STND such that E2-mice consumed less water during day, but more during night, while no effect was detected in cumulative 24 h data (Appendix A).

Locomotor activity was measured using metabolic cages (TSE Systems, Germany) for a 72-h period on D11–D13 and D29–31 during STND and HFD, respectively. A main effect of E2 treatment was present on locomotor activity during 24 h and night. E2 mice on STND were less active during the light phase compared to Veh mice (Figure 2B).

VO_2_ consumption and VCO_2_ production were also measured using metabolic cages. E2 altered VO2 consumption during day and night, and VCO_2_ production during day in longitudinal data. During STND, E2 decreased VO2 consumption during 24 h and day (Figure 2C) and VCO_2_ production at 24 h, day, and night (Figure 2D). Respiratory exchange rate (RER, VO_2_/VCO_2_), a predictor of relative contribution of carbohydrate (value > 0.85) vs lipid (value < 0.8) on energy production, was decreased in E2 group during day, but not during 24 h or night (Appendix A). E2 affected resting energy expenditure (EE) during day and night, with no effect on 24 h data. During STND, EE was attenuated in E2-treated mice during 24 h and day (Figure 2E).

### 2.3. Estradiol Increases Food Intake and Energy Expenditure in Female Mice during HFD

E2 increased HFD consumption during 24 h and night (Figure 3A). Similarly, water intake was increased in E2 mice during 24 h and night (Appendix A). E2 profoundly increased locomotor activity during 24 h and night (Figure 3B). Increased metabolic capacity in E2 mice during HFD was further confirmed by increases in VO2 consumption during 24 h and night and VCO_2_ production, during 24 h and night (Figure 3C,D). Similarly, EE was increased in E2 mice during 24 h and night (Figure 3E). RER was not affected by E2 during HFD feeding (Appendix A).

### 2.4. HFD Increases Body Weight and Fat Mass Gain in Female Mice

To determine the effects of E2 on energy metabolism and gut microbiota under different diet conditions, mice were fed a chow diet (STND) for the first 14 days after OVX and then fed HFD for days 14–45 (Figure 1A). HFD had a profound effect on body weight and fat mass. An interaction between diet and E2 treatment was also present on body weight and fat mass (Figure 1B,C). For the lean mass, while an effect of diet was present on longitudinal data, there was no effect on separate data during STND or HFD (Figure 1D).

### 2.5. HFD Alters Food Intake and Energy Expenditure in Female Mice

A comparison of food intake, in calories, between STND and HFD feeding revealed a main effect of diet during 24 h and day, but not at night. However, there was an interaction of diet and hormone treatment during 24 h, day, and night. Interestingly, Veh mice on HFD ate less calories during 24 h compared to Veh mice on STND. In contrast, E2 mice on HFD ate more calories during 24 h, than during STND (Figure 2 and Figure 3). Both Veh and E2-mice had a greater water intake during STND than HFD, during 24 h and night (Appendix A).

Diet affected locomotor activity during 24 h, day, and night. Within E2 group, mice were more active at 24 h and night following the switch to HFD compared to STND, suggesting that one mechanism by which estrogens prevent HFD-induced obesity is by increasing activity. In contrast, Veh mice were less active during HFD feeding, than on STND, during day. Diet also affected VCO_2_ production (during 24 h, day, and night). Veh mice had reduced VCO_2_ production during HFD than on STND (on 24 h, day, and night data). E2 mice also had reduced VCO_2_ production during HFD compared to STND, but only during night. Moreover, an interaction of E2 and diet was present on VCO_2_ production (on 24 h, day and night data). Similar to VCO_2_ production, Veh mice on HFD had decreased VO2 consumption (during 24 h, day, and night), compared to STND. In contrast, VO2 consumption was increased in E2-mice during HFD (during 24 h, day, and night) (Figure 2 and Figure 3). An interaction of E2 and diet was also present on VO2 consumption (during 24 h, day, and night).

A main effect of diet was also observed on RER, with a lower RER during HFD (during 24 h, day and night) compared to STND. As expected, RER was decreased in both E2 mice (during 24 h data, day, and night) and Veh mice (also during 24 h, day, and night), during HFD feeding due to lipid oxidation. In contrast, EE did not show a main effect of diet, but an interaction of treatment and diet was detected (during 24 h, day, and night). E2-treated mice had an increased EE during HFD (on 24 h, day, and night data) whereas Veh mice during HFD had a decreased EE (during 24 h, day, and night) (Figure 2 and Figure 3).

### 2.6. Estradiol Attenuates Fasting Glucose Levels and Plasma Adipokines in Female Mice

Five-hour fasting blood glucose was measured at different times during STND and HFD, which was lower in E2-treated mice than Veh mice on D8 and D14 (during STND), and D23 (during HFD) (Figure 4A). As a response to changes in plasma glucose and lipids, adipokines are produced, many of which are regulated by E2 [13,43,44]. We investigated if the adipokines, leptin and resistin, are altered by E2 during STND or HFD feeding. Leptin was increased in Veh mice on STND as early as D8 (*p* = 0.029) and on D23 during HFD (*p* < 0.001) (Figure 4B). Compared to E2-treated mice, plasma resistin levels increased in Veh mice during both STND and HFD (Figure 4C). E2 did not alter plasma levels of the pro-inflammatory cytokines, IL-6 and TNF-α (Appendix A). Diet had no effect on plasma glucose and adipokines on the days examined (Figure 4 and Appendix A). Plasma estradiol was measured on D23 of the implant to confirm its release into the circulation, which was significantly higher in E2 group compared to controls (Figure 4D). The intestinal hormones ghrelin and GLP-1 in plasma were undetectable.

### 2.7. Estradiol Improves Insulin Sensitivity in Female Mice on HFD

We performed hyperinsulinemic-euglycemic clamp to measure insulin sensitivity and glucose metabolism in awake mice. E2 mice had an increased glucose infusion rate and increased whole-body glucose turnover compared to Veh controls during clamp (Figure 5A,B). Whole-body glycogen synthesis was increased in E2-treated mice compared to Veh mice (Figure 5C). Insulin-stimulated glucose uptake in skeletal muscle (gastrocnemius) did not differ between E2 and Veh mice, although a trend towards a decrease (*p* = 0.08) was observed in E2 mice, suggesting that skeletal muscle is not primarily responsible for the insulin-stimulated energy utilization in females as an effect of E2 (Figure 5D). E2 did not alter basal or clamp plasma glucose levels, although a trend towards a decrease in basal glucose (*p* = 0.08) was observed in E2 mice (Appendix A). Consistently, a trend towards an increase in basal hepatic glucose production (HGP; *p* = 0.076) was observed in E2 groups, whereas clamp HGP was not affected (Appendix A). Whole-body glycolysis, hepatic insulin action, or liver triglyceride levels were not affected by E2 (Appendix A).

### 2.8. Estradiol Decreases Occludin Expression in Colon in Female Mice Fed HFD

Tight junction proteins provide an indirect measure of intestinal epithelial integrity. Thus, to investigate the role of E2 on healthy epithelial barrier, the tight junction proteins occludin and ZO-1 were measured in female mice after 2 weeks on HFD. Interestingly, E2 treatment reduced the area and intensity of occludin immunoreactivity in the mid-colon compared to Veh mice (Figure 6B,C). There was no effect of E2 on occludin in the proximal and distal colon, or on ZO-1 expression throughout the colon (Appendix A).

### 2.9. Estradiol Alters Gut Microbial Diversity in Female Mice

To identify the effects of E2 on gut microbial diversity during STND or HFD feeding, fresh fecal samples from D1 and D8 (during STND), and from D23 and D42 (during HFD), were analyzed. α-diversity, a measure of within-sample diversity as measured by richness and evenness of species within a population, was not significantly associated with E2 during STND or HFD (Appendix A). β-diversity, a measure of dissimilarity between microbial communities, revealed a distinct clustering of the microbiota communities (Bray-Curtis distance) due to E2 during HFD on D23 (*p* = 0.003) and D42 (*p* = 0.007) (Figure 7B,C, respectively). There were no significant effects of E2 during STND on the aggregate data from D0 and D8 (Figure 7D). These data suggest a profound effect of E2 on gut microbiota diversity in mice fed HFD.

### 2.10. Estradiol Alters Relative Abundances of Gut Microbiota in Female Mice

The generalized mixed effects models with FDR control was used to identify differentially abundant taxa (q-value < 0.05). A total of 14 taxa differed between E2 and Veh groups (Figure 8A,B). Of these 14 taxa, Verrucomicrobia (phylum) and all its lower taxa levels, including Verrucomicrobiae (order), Verrucomicrobiales (class), Verrucomicrobiaceae (family), and the genus *Akkermansia,* were increased in E2 mice, with the most pronounced differences during HFD feeding (Figure 8B). *Dorea* spp. were also increased in E2 mice compared to Veh mice (Figure 8A,B).

Other taxa, including Erysipelotrichi (order) and its genus *Coprobacillus,* and Streptococceae (family) and its genus *Lactococcus* were decreased in E2 mice compared to Veh controls during HFD feeding. In addition, the family Clostridiaceae and its genus *Clostridium,* were decreased in E2 mice both during STND and HFD feeding (Figure 8B).

### 2.11. HFD Alters Gut Microbiota Diversity and Relative Abundances in Female Mice

Similar to previous reports mostly in males [31,32,45,46,47,48], HFD profoundly affected gut microbiota composition in female mice. HFD decreased microbiota richness (Chao1; *p* = 0.002; Appendix A) and increased evenness, the measure of homogeneity of species distribution in a population (Pielou’s; *p* < 0.001; Appendix A).

Diet also profoundly altered microbiota community structures. The microbial communities distinctly clustered between STND and HFD feeding (*p* < 0.001, PERMANOVA), as depicted by the PC1 (64.3%; Figure 7A). The effect of E2 was strong during HFD (Figure 7B), while no effects of E2 were detected during STND (Figure 7C). Moreover, a total of 49 taxa were differentially associated with STND vs HFD, of which 39 were positively associated with HFD, while only 10 were positively associated with STND, further supporting a profound effect of HFD on gut microbiota (Figure 8C,D).

HFD increased 39 taxa including Firmicutes and its lower taxa Clostridia (order), Clostridiales (class), the families Mogibacteriaceae and Peptostreptococcaceae, and the genera *Dorea, Ruminococcus, Anaerotruncus,* and *Oscillospira.* HFD increased additional taxa within the Firmicutes, including Erysipelotrichi (order) and its lower taxa Erysipelotrichales (class), Erysipelotrichaceae (family) and *Allobaculum* and *Coprobacillus*. Two other families, Bacteroidaceae and Streptococcaceae, and their genera *Bacteroides* and *Lactococcus,* respectively, were also positively associated with HFD. Furthermore, HFD increased Proteobacteria and its lower taxa, including Deltaproteobacteria (order), Desulfovibrionales (class), Desulfovibrionaceae (family), and *Desulfovibrio.* In addition, the phylum Verrucomicrobia and all of its lower taxa levels, including Verrucomicrobiae (order), Verrucomicrobiales (class), Verrucomicrobiaceae (family), and the genus *Akkermansia,* were increased during HFD feeding compared to STND. Actinobacteria (phylum) and its lower taxa at all levels, including Coriobacteriia (order), Coriobacteriales (class), Coriobacteriaceae (family) and its genus *Adlercreutzia,* were also increased as a result of HFD feeding.

Ten taxa were increased during STND compared to HFD, including Turibacteriales (class), its family Turibacteriaceae and the genus *Turibacter*, as reported previously [49]. The relative abundances of the family Clostridiaceae and its genus *Clostridium* and *Coprococcus* were also increased during STND compared to HFD. Similarly, Tenericutes (phylum), its lower taxa Mollicutes (order), and the genus *RF39* were increased during STND.

### 2.12. Gut Microbiota Associates with Metabolic Status in Female Mice

To investigate if metabolic changes are associated with changes in the gut microbiota community, correlation analysis was performed between the measures. PERMANOVA test based on Bray-Curtis distance followed by FDR correction (q-value < 0.1) revealed significant correlations of body weight (q = 0.015), plasma glucose (q = 0.025) and physical activity (q = 0.09) with microbial community distances. To identify specific taxa that are linked to E2-dependent metabolic effects, correlation analysis was done between the microbial taxa that were altered by E2 treatment and the major metabolic profiles (Figure 9). Verrucomicrobia, along with its lower taxa levels, including *Akkermansia*, negatively correlated with body weight, fat mass, and leptin, suggesting *Akkermansia* as a microbial mediator of E2-dependent protection against obesity. Verrucomicrobiae and *Dorea*, both increased in E2 mice, were negatively associated with blood glucose levels. In addition, *Dorea* was positively associated with physical activity. In contrast, some taxa that were increased in Veh mice, including Streptococcaceae and its genus *Lactococcus*, were positively associated with body weight, fat mass, and leptin, suggesting these taxa are predictors of obesity. Similarly, Erysipelotrichi, its family Erysipelotrichaceae and genus *Coprobacillus*, were positively associated with body weight. Interestingly, *Coprobacillus* was positively correlated with fat mass, but negatively correlated with physical activity and basal energy expenditure, suggesting a negative impact of this microbe on metabolic health in female mice.

## 3. Discussion

In the current study, we investigated the comprehensive mechanisms by which estrogens protect females against diet-induced obesity and insulin resistance. E2 treatment prevented HFD-induced weight gain and adiposity in ovariectomized adult mice, consistent with earlier work from our group and others [11,13,14,42,44]. The E2-dependent protection against HFD-induced obesity was most strongly associated with increased physical activity and basal energy expenditure. E2 prevented hyperglycemia during both STND and HFD intake. Consistent with previous studies, E2 decreased the plasma adipokines leptin and resistin [13,44]. Furthermore, hyperinsulinemic-euglycemic clamp results showed that E2 improved systemic insulin sensitivity and glucose turnover in HFD-fed mice. However, skeletal muscle glucose uptake, hepatic glucose production, and hepatic triglycerides were not altered by E2 in HFD-fed mice. These data demonstrate tissue-specific effects of E2 in providing the protective mechanisms against HFD-induced obesity and insulin resistance.

Estrogen receptors (ER) exist in two forms, ERα and ERβ, which are transcribed from different genes [50,51]. These subtypes differ in their abilities to bind different ligands, are expressed differently in specific tissues and mediate different functions in behavior and physiology [51,52]. Intestinal epithelium predominantly expresses ERβ [53]. To identify any effects of estrogens in this key metabolic passageway, we analyzed changes in the intestinal epithelium in mice with or without E2. The tight junction protein, occludin, was decreased in the colon of E2-treated mice fed HFD, suggesting that HFD-induced increase in gut permeability, due to the depletion of tight junction proteins and mucus layer thickness [54,55], is modulated by E2. In future work, it will be important to study additional tight junction proteins combined with in vivo gut permeability assays to further explore the effects of E2 and diet on gut integrity and barrier function.

Host metabolic status can be predicted by its gut microbiota community and composition. Metabolic syndrome, characterized by adiposity, hyperlipidemia and hyperglycemia, is linked to dysbiosis of the gut microbial ecosystem [56,57,58]. However, we currently lack a full understanding of the parallel assessment of gut microbiota and metabolic changes within the same animals as an effect of E2, which limits the knowledge of any direct interactions between the host metabolic status and microbial factors. Thus, in the present study, we assessed if changes in gut microbiota are linked to the protective effects of estrogens against obesity, hyperglycemia, and insulin resistance. E2 altered microbial communities and taxa, with a profound effect during HFD feeding. Notably, the relative abundances of the phylum Verrucomicrobia, including its major constituent genus *Akkermansia, and Dorea* (phylum Firmicutes), were significantly increased by E2 during HFD. An increase in *Akkermansia* abundance is also associated with E2-mediated protection against western diet-induced obesity and metabolic syndrome in ovariectomized mice [41]. In the current study, we further identified an association between *Akkermansia* and multiple metabolic measures. *Akkermansia* negatively correlated with body weight and fat mass, suggesting it functions in the protective effects of E2 on metabolic health. Similarly, *Dorea* was positively associated with physical activity. In support of these findings, ovariectomy increases weight gain in both STND- and HFD-fed rats and is associated with changes in gut microbiota [59]. Similarly, in a PCOS mouse model, FMT from androgen-treated mice disrupts metabolic and endocrine health in germ-free recipients, whereas gut microbiota from control donors protects against metabolic dysregulation [60,61,62]. In a different study, diet-independent, ovariectomy-induced weight gain was not rescued by cohousing with intact mice, with the goal of transferring of gut microbiota [49]. It is possible that a more complete transfer of microbiota is needed to rescue this ovariectomy-induced weight gain, such as fecal microbiota transfer (FMT) by gavage or co-housing combined with FMT. Nevertheless, the present findings, taken together with previous ones, suggest that gut microbiota functions in metabolic dysregulation caused by diet or sex hormones.

The relative abundance of *Akkermansia,* the only intestinal resident genus of the phylum Verrucomicrobia, was significantly increased in E2-treated mice. *Akkermansia,* a mucin-degrader and a producer of short chain fatty acids [63,64,65,66], is decreased in obese humans, including obese pregnant women [65,67,68,69]. In postmenopausal women, *Akkermansia* is negatively correlated with insulin resistance and dyslipidemia [70]. Similarly, in ovariectomized mice fed a western diet, *Akkermansia* was increased following E2 treatment [41]. Administration of heat-killed *Akkermansia muciniphila* decreased body weight, fat mass, and hip circumference in obese women, highlighting its beneficial role in women’s metabolic health [71]. In further support, *A. muciniphila* supplementation in male mice attenuated HFD-induced obesity and inflammation and improved insulin signaling [72,73,74,75]. In the present study, the relative abundance of *Dorea* was also increased by E2 treatment in HFD-fed females. Similar to *Akkermansia*, *Dorea* is a mucin degrader, suggesting these two microbes are co-altered in response to changes in diet or hormones, likely due to the similar nutrient environment and/or quorum sensing [76]. Taken together, these findings suggest that *Akkermansia* and *Dorea* contribute to the E2-mediated compensatory protection against HFD-induced metabolic changes.

*Coprobacillus*, *Lactococcus*, and *Clostridium*, including their families Erysipelotrichaceae, Streptococceae and Clostridiaceae, respectively, were increased in Veh mice. Among these microbes, *Coprobacillus* and *Lactococcus* were positively correlated with body weight and fat mass, suggesting they contribute to obesity in E2-deficient female mice. An inverse correlation of *Lactococcus* and *Coprobacillus* with E2 has been previously demonstrated in female mice on standard diet [77] and female *ob/ob* mice on HFD [42]. *Lactococcus* are efficient energy harvesters through the conversion of glucose to pyruvate [78]. *Coprobacillus* produce β-galactosidases, enzymes necessary for the breakdown of galactosides, such as lactose in food [79]. These and other gut microbes can also affect intestinal endocrine cells through metabolite production [80,81], which can impact the development of type 2 diabetes and obesity [82]. In future studies, it will be important to determine if selective depletion of these microbes mitigates the metabolic insult caused by the loss of estrogens in females.

Intake of a high-calorie diet during menopause, a period characterized by a slowed metabolic rate, further increases the risk of obesity and metabolic disorders in women [4,7,9]. In the current study, the protective effect of E2 treatment on metabolic status was profound during HFD intake in female mice, which is consistent with previous reports [11,13,14,32,42,57,83,84,85]. The increases in body weight and fat mass, and a decrease in basal energy expenditure, due to HFD feeding were attenuated by E2. E2 corrected HFD-induced positive energy balance primarily by increasing basal energy expenditure and locomotor activity, extending previous findings [14]. Moreover, E2 increased energy utilization in HFD-mice by increasing systemic insulin sensitivity and whole-body glucose turnover. Since these effects were not associated with increased muscle glucose metabolism, other estrogen-sensitive organs might be responsible for increased glucose utilization in E2-treated mice. E2-mediated improvements in some measures of insulin sensitivity have also been demonstrated previously [14,44]. Given that mice lacking ERα are insulin resistant [86], these effects of E2 on metabolic pathways discussed above are most likely mediated by ERα.

The present and previous studies have found that levels of the adipokines, leptin and resistin, were decreased by E2 in female mice [13,44]. The present study reveals that this decrease in leptin levels was associated with gut microbiota. In particular, leptin was negatively associated with *Akkermansia,* a positive microbial predictor of metabolic health, whereas was positively associated with *Lactococcus* [42]. Leptin decreases food intake and increases energy expenditure [87,88]. However, increased circulating leptin is positively linked to metabolic syndrome in women [89,90]. In addition, resistin deficiency is associated with increased insulin sensitivity, particularly through a reduction in hepatic glucose production [91,92]. Therefore, the early changes in adipokines observed in the present and previous studies [13,44] may serve as early markers of diet-induced obesity and insulin resistance, as well as measures of the E2 response against various metabolic insults. Moreover, the E2-dependent downregulation of leptin and its interaction with gut microbiota may provide an essential braking mechanism against the development of diet-induced obesity.

## 4. Materials and Methods

The following animal experiments were performed at the University of Massachusetts Medical School (PROTO202000104, 11/01/20). All procedures were approved by the Institutional Animal Care and Use Committees of UMass Medical School and Wellesley College and performed in accordance with National Institutes of Health Animal Care and Use Guidelines.

### 4.1. Diet and Ovariectomy

Female C57BL/6J mice were purchased from The Jackson Laboratory and housed in the animal facility at UMass Medical School. Ten-week-old female C57BL/6J mice were bilaterally ovariectomized (OVX) and silastic capsules filled with 17β-estradiol (E2, 50 μg in 25 μL of 5% ETOH/sesame oil, *n* = 6) or vehicle (Veh, *n* = 6) were implanted subcutaneously [11,42]. Mice were singly housed and fed a chow diet (STND; 13.5% calories from fat, #5001, Purina, LabDiet, Fort Worth, TX, USA) for the first 14 days after OVX. To test the effects of E2 on metabolism and gut microbiota under HFD, mice were put on a HFD containing 60% kcal fat (#D12492, Research Diets, New Brunswick, NJ, USA) for the remainder of the study (days 14–45; Figure 1A).

### 4.2. In Vivo Assessment of Energy Balance Using Metabolic Cages

We performed a 3-day measurement of energy balance (i.e., food intake, VO_2_ consumption and VCO_2_ production, energy expenditure, respiratory exchange ratios, and physical activity) using metabolic Cages (TSE Systems, Germany) in mice (*n* = 6 per treatment group) on D11-13 during STND and D29-31 during HFD as described previously [93,94]. The O_2_ consumption and CO_2_ production were used to calculate the respiratory exchange ratio (RER). The horizontal and vertical movement (XYZ-axis) were measured in the cages as an index of locomotor activity. Body composition (fat/lean mass) was assessed by proton magnetic resonance spectroscopy (1-H MRS; EchoMRI, Houston, TX, USA) once each week (Figure 1A).

### 4.3. Measurement of Glucose Metabolism Using Hyperinsulinemic-Euglycemic Clamp

On days 37–39, anesthetized mice underwent a survival surgery to establish an indwelling catheter in the jugular vein. One week after the surgery, following overnight fasting, a 2-h hyperinsulinemic-euglycemic clamp was conducted in awake mice with a primed (150 mU/kg body weight) and continuous infusion of human insulin at a rate of 2.5 mU/kg/min to raise plasma insulin within a physiological range [95]. D-[3-3H] glucose was intravenously infused using microdialysis pumps during the experiments to assess the whole-body glucose turnover [96]. Blood samples were collected at 10–20 min intervals for the immediate measurement of plasma glucose, and 20% glucose was infused at variable rates to maintain euglycemia. To estimate insulin-stimulated glucose uptake in individual organs, 2-[1-14C] deoxy-D-glucose (2-[14C] DG) was administered as a bolus (10 µCi) at 75 min after the start of clamp. Blood samples were taken for the measurement of plasma [3H] glucose, 3H2O, and 2-[14C] DG concentrations. At the end of the clamp, mice were anesthetized and tissue samples were taken for biochemical and molecular analyses.

### 4.4. Calculation of In Vivo Glucose Metabolism

Basal whole-body glucose turnover was determined as the ratio of the [3H] glucose infusion rate to the specific activity of plasma glucose at the end of basal period, as previously described [96]. Insulin-stimulated whole-body glucose uptake was determined as the ratio of the [3H] glucose infusion rate to the specific activity of plasma glucose during the final 30 min of clamps. Hepatic glucose production during insulin-stimulated state (clamp) was determined by subtracting the glucose infusion rate from the whole-body glucose uptake. Whole-body glycolysis was calculated from the rate of increase in plasma 3H2O concentration from 90–120 min of clamp. Whole-body glycogen plus lipid synthesis was estimated by subtracting whole-body glycolysis from whole-body glucose uptake. Since 2-DG is a non-metabolizable glucose analog, insulin-stimulated glucose uptake in skeletal muscle were estimated by determining muscle-specific content of 2-[14C] DG-6-P. Skeletal muscle glucose were calculated from plasma 2-[14C] DG decay profile and intracellular 2-[14C] DG-6-P content.

### 4.5. Biochemical Assays

Blood samples were collected after 5-h fasting on day (D) D8 (during STND) and on D23 (9 days on the start of HFD) by tail vein puncture (Analytic Core, MMPC). Plasma E2 on D23 was measured using Mouse/Rat Estradiol ELISA kit (#ES180S, Calbiotech [12,97]. The cytokines Il-6 and TNF-α, the adipokines leptin and resistin, and intestinal hormones, ghrelin and GLP-1 were measured using an ELISA with a Luminex 200 Multiplex system (Millipore, Darmstadt, Germany).

Glucose concentrations during clamps were analyzed using clinical glucose analyzer, and insulin levels were measured using an ELISA kit. Plasma [3H] glucose, 2-[14C] DG, and 3H_2_O concentrations were determined following deproteinization of samples using liquid scintillation counter on dual channels for separation of 3H and 14C. The radioactivity of 3H in tissue glycogen was determined by precipitating glycogen with ethanol. Organ-specific 2-[14C] DG-6-phosphate concentrations were determined using ion-exchange column as previously described [98]. Hepatic intracellular triglyceride level was measured using spectrophotometry using triglyceride assay kit after digesting tissue samples in chloroform-methanol.

The following animal experiments were performed at Wellesley College, and all procedures were approved by the Institutional Animal Care and Use Committees of Wellesley College (#2101, 02/05/21) and performed in accordance with National Institutes of Health Animal Care and Use Guidelines.

### 4.6. Fecal DNA Extraction and Sequencing

DNA was extracted from fresh frozen fecal samples on D0 and D8, during STND and D23 and D42, during HFD, using MO BIO PowerSoil DNA Isolation Kit (Valencia, CA) with minor adjustments to the manufacturer’s protocol, as described previously [42]. The DNA quality and quantity were assessed using Nanodrop spectrophotometer (Thermo Scientific, Waltham, MA, USA). 16S rDNA was amplified at the V3-V4 region using the universal 16S rDNA primers: for- ward 341F (5′-CCTACGGGAGGCAGCAG-3′) and reverse 806R (5′-GGACTACHVGGGTWTCTAAT-3′) with sequence adapters on both primers and sample-specific Golay barcodes on the reverse primer [99]. The amplicons were quantified by PicoGreen (Invitrogen, Carlsbad, CA, USA) and pooled in equal concentrations. The pooled amplicons were cleaned using UltraClean PCR Clean-Up Kit (MO BIO, Carlsbad, CA, USA) followed by quantification using the Qubit (Invitrogen, Carlsbad, CA, USA).

Samples were multiplexed and paired-end sequenced using 16S rDNA primers on an Illumina MiSeq (Illumina, San Diego, CA, USA) at the Microbiome Core (Mayo Clinic, Rochester, Minnesota). Paired R1 and R2 sequence reads were processed via the hybrid-denovo bioinformatics pipeline, which clustered a mixture of good-quality paired-end and single-end reads into operational taxonomic units (OTUs) at 97% similarity level [100]. OTUs were assigned taxonomy using the RDP classifier trained on the GreenGenes database (v13.5) [101,102]. A phylogenetic tree based on FastTree algorithm was constructed based on the OTU representative sequences [103]. The total number of reads ranged from 44,869 to 697,323 with a median of 122,445 reads per sample.

### 4.7. Intestinal Tissue Processing for Histology

Mice used for the intestinal histology analysis were housed in the animal facility at Wellesley College. Mice were ovariectomized and implanted with E2 (*n* = 6) or Veh (*n* = 6), as described above. Animals were fed STND for 7 days and switched to HFD containing 60% kcal fat (#D12492, Research Diets) on day 8 (D8). On D22 of OVX (after 2 wks on HFD), mice were euthanized and colons were collected.

Mice treated with E2 or Veh (*n* = 6/group) were euthanized after 2 wks on HFD to investigate the effects of E2 on intestinal epithelium. Colon was prepared as previously described, with some modifications [104]. In brief, colon tissue was longitudinally cut open and vigorously washed 3 times in 1X PBS (pH 7.2). The colon tissue was then dipped in modified Bouin fixative (50% EtOH in 5% Acetic acid in 1X PBS) for 5 min. The tissue was rolled on a toothpick, transferred to a tissue cassette, and fixed overnight in 10% formalin at room temperature. Prior to paraffin embedding, tissue was processed using a tissue processor (CITADEL 2000, Thermo Fisher) at DERC Morphology Core, UMass Medical School. Briefly, the tissue was incubated in 70% EtOH for 1x, 95% EtOH for 1x, and 100% EtOH and 3x, for 1 h each on an orbital shaker. The tissue was then incubated in xylene (#X5SK, Thermo Fisher) 3x for 1 h each and kept in a tissue mold and incubated in paraffin (Histoplast IM, Cat# 8331, Thermo Fisher, Waltham, MA, USA) at 58 °C 2x for 2 h. The tissue roll was sectioned at 5 μm thickness on a cryostat.

### 4.8. Triple-Label Immunohistochemistry for Tight Junction Proteins

Triple-label immunohistochemistry was done on colon sections to quantify tight junction proteins, occludin and zona occludens 1 (ZO-1), and a nucleic acid stain (DAPI) in colonic epithelium including crypts as described previously [105]. In brief, the colon sections were deparaffinized in xylene and dehydrated in 100% ethanol, followed by rehydrating in graded ethanol concentrations of 95%, 70% and 50%. Antigen retrieval was done to increase the antigen accessibility by incubating the slides for 30 min in Tris-EDTA (pH 9.0) in boiling water. Slides were allowed to cool and washed in 0.5% sodium borohydride (*w/v*) in TBS for 20 min to remove excess fixative. Following additional washes in TBS, the sections were incubated in blocking buffer (10% normal donkey serum, 0.3% Triton, 1% BSA) for 30 min. The sections were then incubated at 4 °C overnight in rabbit polyclonal antibody directed against human occludin (1:100, #ab168986, Abcam) [106] and goat polyclonal antibody directed against C-terminal of human ZO-1 (1:100; #ab190085, Abcam, Cambridge, MA, USA) [107]. The specificities of occludin and ZO-1 on mouse intestinal tissue have been verified previously [74,108]. The following day, sections were washed then incubated for 1 h in dark at room temperature with a nucleic acid stain, DAPI, at a concentration of 30 uM and fluorescently labeled donkey-anti-rabbit (1:100; Alexa Fluor 647, Invitrogen) [109] and donkey-anti-goat (1:100; Alexa Fluor 488, Invitrogen, Waltham, MA, USA) [110] secondary antibodies for the detection of occludin and ZO-1, respectively. Slides were washed, coverslipped with Fluorogel (Electron Microscopy Sciences, Hatfield, PA, USA), stored overnight in dark, and imaged within two days.

### 4.9. Imaging by Confocal Microscopy and Analysis

The proximal, middle and distal regions of the colon were imaged using a Leica laser scanning confocal microscope (TCS SP5 II), equipped with 405 Diode, Argon, HeNe 594, and HeNe 633 lasers and with Leica software (LAS version 2.7.3.9) [111]. All images were taken under 200x magnification with the PL APO dry-objective (numerical aperture, 0.7). The gain and offset values for each laser were optimized for each channel and kept constant for all animals. Sections of 1 µm thickness were optically imaged and analyzed using the NIH ImageJ software (version 1.52) [112]. A representative section per animal per subregion was analyzed using uniform regions of interest (ROI), which were kept constant for each subregion across all animals. For the quantification of immunolabeling, threshold for each laser channel was set separately based on a scale of 0–255, to minimize the background. Using three random images per treatment, the threshold that displayed the immunolabeling signal closest to the unprocessed original image was chosen and applied across all animals. Any value below the threshold was considered to be background. The % area with labeling above threshold and the mean pixel intensity were collected within the selected ROI for each subregion.

### 4.10. Statistical Analysis

#### 4.10.1. Metabolic Data

The effects of diet and E2 treatment were analyzed on longitudinal data using mixed model repeated measures (“lme” in R “nlme” package) [113] using mouse as a random subject. Once the main effects were observed, the separate effects during STND and HFD were measured using repeated measures ANOVA (spss v.24) [114]. Data from metabolic cages, including food and water intake, locomotor activity, and respiration (O_2_ consumption and CO_2_ production) and its derivative measures, respiratory exchange ratio and resting energy expenditure, were recorded over 72 h and averaged to produce 24-h data, and analyzed using the *t*-test. Fasting blood glucose, plasma hormones and cytokines, and end point measures (including clamp data) were analyzed using the *t*-test. *p* < 0.05 was considered statistically significant.

#### 4.10.2. 16S rRNA Sequence Data

The data were rarefied to the minimum depth of 44,869 prior to the α-diversity and β-diversity analyses [115]. For α-diversity analysis (Chao1 richness and Pielou’s evenness indices), linear mixed effects models were fitted to the alpha diversity measures with a random intercept for each mouse (“lme” in R “nlme” package). Wald test was used for assessing the significance. For β-diversity analysis (Bray-Curtis distance), PERMANOVA test (R adonis, 1000 permutations) was used to test whether overall microbiota composition is associated with E2 or diet. For testing the E2 effects, the mouse (not individual sample) was the permutation unit; for testing the diet effects, the mouse was the permutation stratum (i.e., permutation only occurred within the same mice) [116]. The R^2^ was given as the effect size.

Differential abundance analysis of treatment (E2 vs Veh) and diet (HFD vs STND) effect was performed on the phylum, class, order, family and genus level. Only taxa with prevalence >10% and maximum proportion >0.2% were tested. Generalized linear mixed effects model (R “glmmPQL” function, over-dispersed Poisson regression, random intercept) was fitted to the aggregated counts accounting for within-mouse correlation [117]. The library size was estimated using GMPR method [118]. The log library size was included as an offset in the regression model. Treatment and diet variables were included as covariates. Potential treatment and diet interaction (GxD) was also investigated by including the interaction term in the regression model. Wald test was used to test the significance of the association. Data were winsorized at 95% quantile (i.e., we replace outlier counts with 95% quantile) to reduce the influence of potential outliers. False discovery rate (FDR) control (BH procedure, R p.adjust function) was used for multiple testing correction and performed on each taxonomic level from phylum down to genus. The taxa with an FDR-adjusted *p* value (or q value) < 0.05 were considered as statistically significant.

#### 4.10.3. Correlation Analysis of Microbiome and Metabolic Data

To identify if any changes in E2-dependent metabolic effects significantly associate with changes in gut microbiota, correlation analyses were performed between the two outcomes. PERMANOVA was used to perform an overall association test based on the Bray-Curtis distance. For metabolic measures where multiple samples within the same mouse were obtained, within-mouse permutations were done. Next, correlation tests were done to identify microbial taxa associated with metabolic measures. To control for the potential confounding effects due to diet and E2 treatment, residuals were taken by fitting regression models (linear mixed effects model) to the microbial taxa abundance (square-root transformed) and metabolic measures adjusting for diet and E2 effects. Spearman correlation tests were then performed on the residuals. To reduce multiple testing burden, correlation analyses were focused on the taxa associated with E2 treatment. The associations with an FDR-adjusted *p* value (or q value) < 0.1 were considered as statistically significant.

## 5. Conclusions

The present study provides compelling evidence that estrogens profoundly impact energy and metabolic homeostasis in female mice. Consistent with previous studies, the key metabolic changes, including food intake, energy expenditure, and glucose turnover, were improved by E2 in females fed HFD. Moreover, the present findings reveal that gut microbiota and gut barrier integrity are additional targets of E2-mediated protection against diet-induced metabolic disorders. Furthermore, the role of gut microbiota in metabolic health is supported by the present findings of strong correlations of multiple microbial taxa with specific metabolic measures and physical activity. In future studies, it will be important to perform shotgun metagenomics for the functional study of the gut microbiome and explore the potential beneficial effects of *Akkermansia* and other microbes identified in this study and their causative links with metabolic protection in females provided by E2. In addition, identification and characterization of microbial metabolites that contribute to the beneficial effects of E2 on metabolism will provide important insights for targeting gut microbiota to improve women’s metabolic health.

## Figures and Tables

**Figure 1 metabolites-11-00499-f001:**
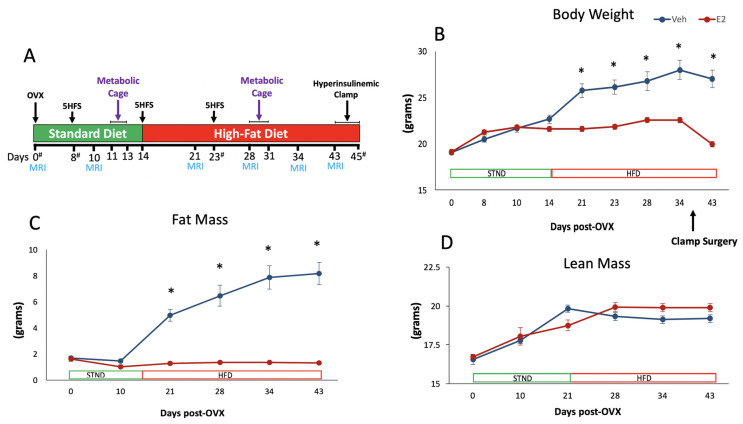
Estradiol attenuates weight gain in female mice on a high-fat diet (HFD). Experimental timeline (**A**). Ten-week old mice were ovariectomized and implanted with capsules containing E2 (50 µg) or Veh implants (*n* = 6/group). Animals were placed in metabolic cages for 3 days, once each during STND and HFD. Body weight (**B**) Fat mass (**C**) Lean mass (**D**). Mice were switched from standard diet (STND) to HFD on day 14. Surgery for the hyperinsulinemic- euglycemic clamp on days 37–39 (4 mice/day) resulted in weight change in both groups. Error bars are shown as ± SEM. * denotes *p* < 0.001, using repeated measures ANOVA followed by *t*-test. OVX: ovariectomy; 5HFS: 5-h fasting blood glucose; MRI: body composition measurement using proton magnetic resonance spectroscopy (1H-MRS); Clamp: hyperinsulinemic-euglycemic clamp; # indicates fecal sample collection days.

**Figure 2 metabolites-11-00499-f002:**
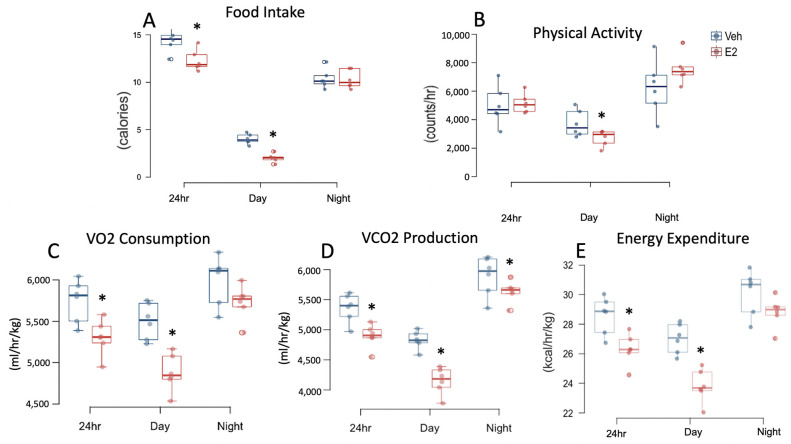
Estradiol decreases food intake and energy expenditure in female mice fed a STND. Data for Food intake (**A**) Physical activity (**B**) VO2 consumption (**C**) VCO_2_ production (**D**) Resting energy expenditure (**E**) were collected from mice in metabolic cages on days 11–13. The average 24-h data were obtained from 72-h data and used for statistical analysis. * indicates differences between E2 and Veh mice (*n* = 6/group) (*p* < 0.05; *t*-test).

**Figure 3 metabolites-11-00499-f003:**
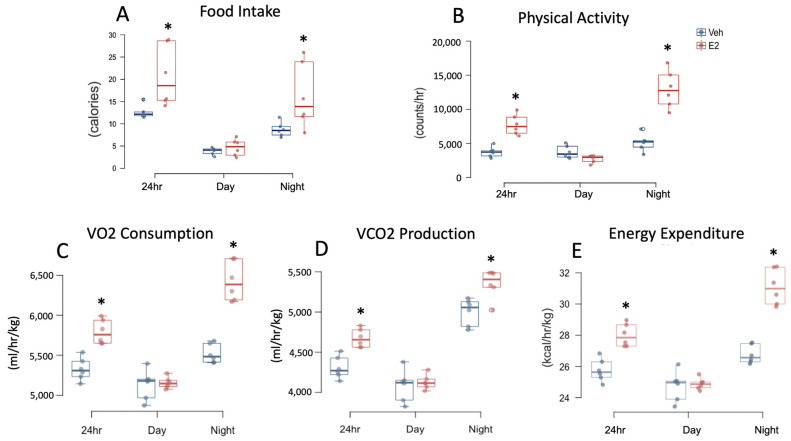
Estradiol increases food intake, water intake, and energy expenditure in female mice fed HFD. Food intake (**A**) Physical activity (**B**) VO2 consumption (**C**) VCO_2_ production (**D**) Resting energy expenditure (**E**) were measured in mice in metabolic cages on days 15–17 of HFD feeding. The average 24-h data were obtained from 72-h data and used for statistical analysis. * indicates differences between E2 and Veh mice (*n* = 6/group) (*p* < 0.05; *t*-test).

**Figure 4 metabolites-11-00499-f004:**
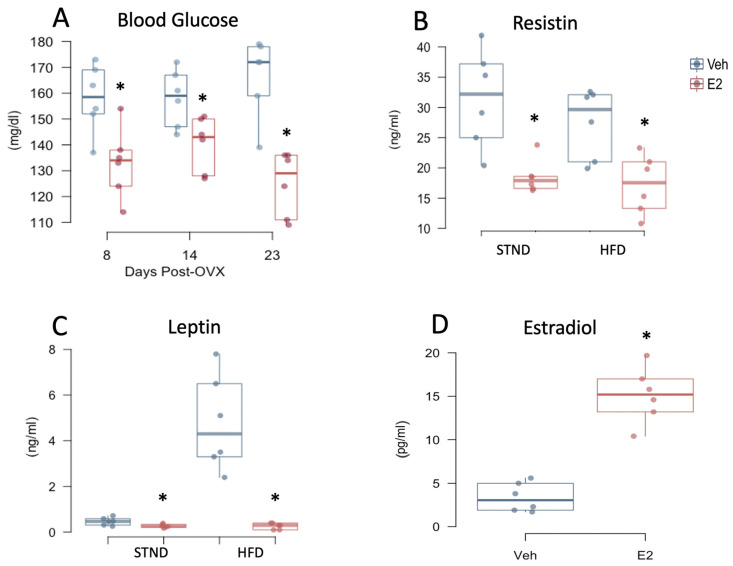
Estradiol decreases plasma glucose and adipokines in female mice independent of diet. 5 h-fasting blood glucose (**A**) on days 8 and 14, both during STND and on day 23, during HFD. Resistin (**B**) and leptin (**C**) were measured on D8 during STND and on D23 during HFD. Plasma estradiol levels were measured on D23 to confirm physiological levels in the treatment group (**D**). * indicates differences between E2 and Veh mice (*n* = 6/group) (*p* < 0.05, repeated measures ANOVA followed by *t*-test for A, B, and C, and *t*-test for D).

**Figure 5 metabolites-11-00499-f005:**
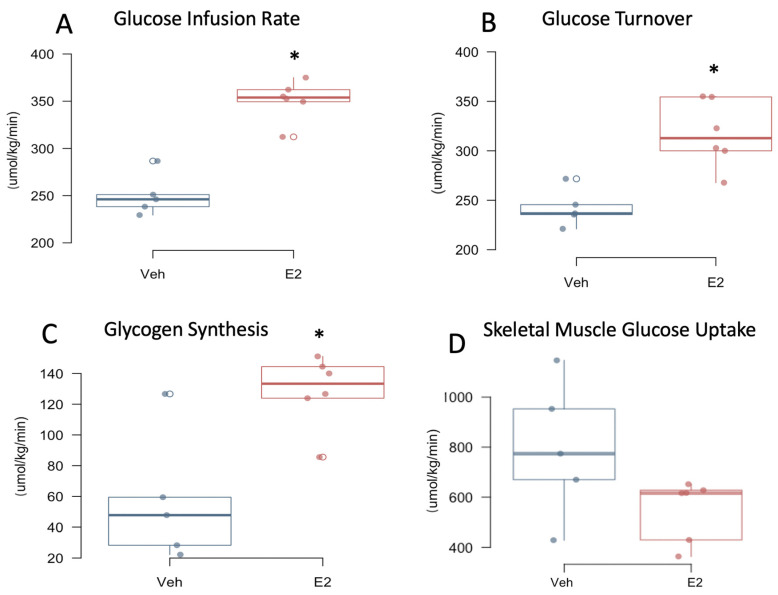
Estradiol increases insulin sensitivity and glucose utilization in female mice on HFD. Mice underwent hyperinsulinemic-euglycemic clamp on days 43–45, a week following jugular vein surgery. Glucose infusion rate (**A**) Glucose turnover (**B**) Glycogen synthesis (**C**) Skeletal muscle glucose uptake (**D**). * indicates differences between E2 (*n* = 6) and Veh (*n* = 5) mice * (*p* < 0.05, *t*-test).

**Figure 6 metabolites-11-00499-f006:**
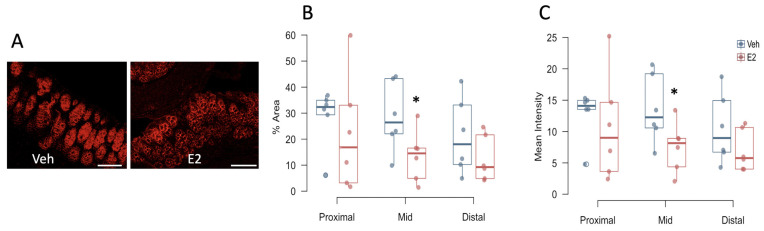
Estradiol decreases occludin immunoreactivity in colonic epithelium in female mice fed HFD. Colon tissues were collected on day 22 (15 days after start of HFD). Occludin immunolabeling in representative mid-colon sections from a Veh (**left**) and an E2-treated (**right**) mouse (**A**), Percent area (**B**) and mean intensity (**C**) of occludin immunoreactivity in the three subdivisions of the colon. Scale bar = 100 μm. * indicates differences between E2 and Veh mice (*n* = 6/group) (*p* < 0.05, *t*-test).

**Figure 7 metabolites-11-00499-f007:**
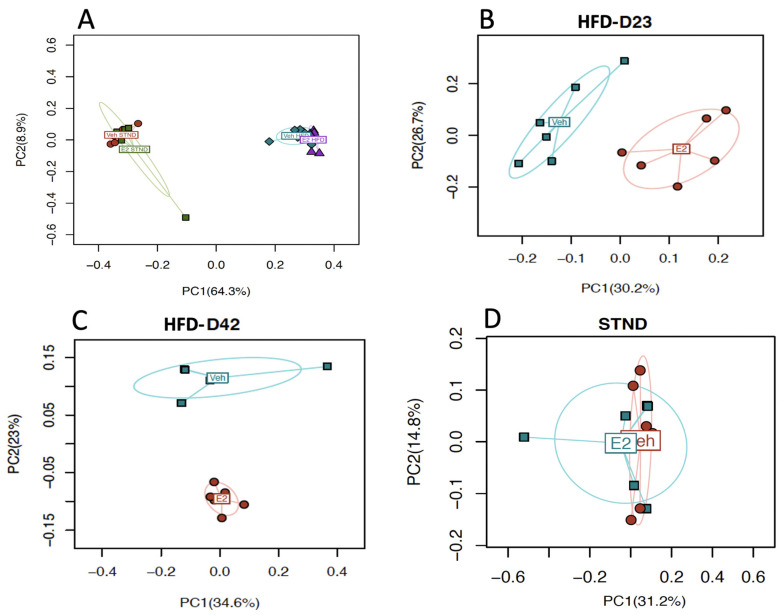
Estradiol and HFD alter gut microbiota β-diversity in female mice. Bray-Curtis distance between aggregate microbiota communities shows a distinct clustering between STND and HFD (**A**) and between E2 and Veh-treated animals during HFD, on Day 23 (**B**) and Day 42 (**C**), but not during STND (**D**). For A, (*n* = 12/group) and for B-D, (*n* = 6/group). *p* < 0.05 (PERMANOVA) considered significant.

**Figure 8 metabolites-11-00499-f008:**
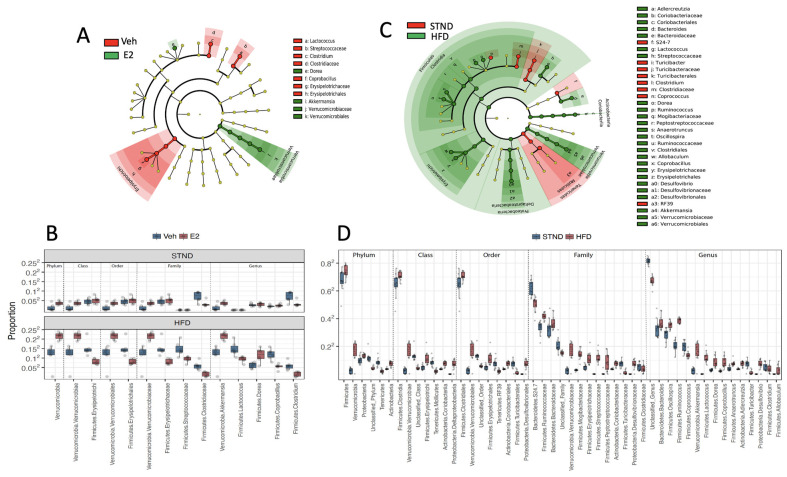
Estradiol and HFD alter gut microbiota taxa diversity in female mice. Cladogram (**A**) and relative proportions (**B**) of taxa associated with E2 or Veh, subgrouped as phylum, class, order, family and genus. Samples during STND (D0 and D8) and HFD feeding (D23 and D45) were analyzed. Cladogram (**C**) and relative proportions (**D**) of taxa associated with STND (D0 and D8) or HFD (D23 and D45). The innermost nodes in the cladogram represent phyla and the connecting outer nodes represent lower taxa within the phylum. q-value < 0.05 (generalized mixed effects model) considered significant. *n* = 24/group for (**A**,**C**), and *n* = 12/group for (**B**,**D**).

**Figure 9 metabolites-11-00499-f009:**
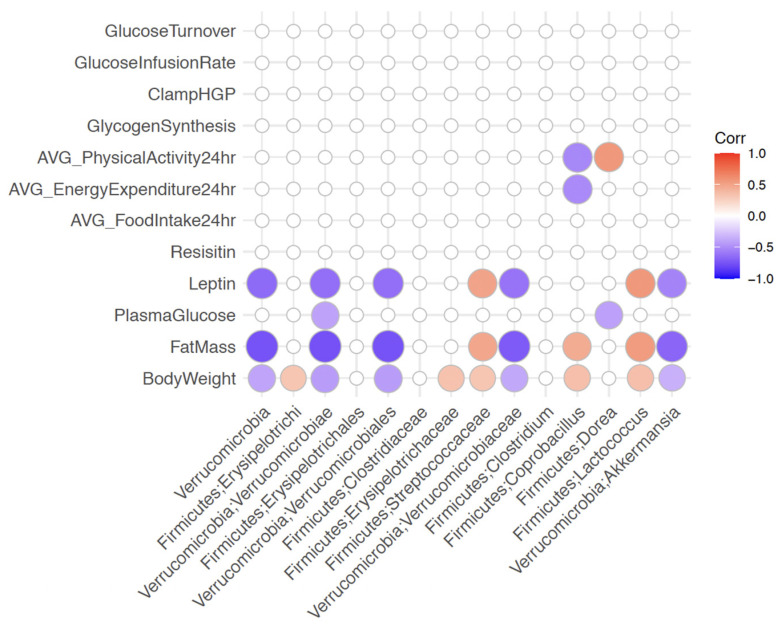
Metabolic changes correlate with multiple gut microbiota taxa in female mice. Spearman correlation tests were done on the taxa associated with E2 treatment. Red circles represent positive correlation whereas purple circles represent negative correlation, with increasing size of circles indicating stronger associations. q-value < 0.1 considered significant.

## Data Availability

Datasets generated during and/or analyzed during the current study are not publicly available but can be made available upon reasonable request.

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
