# Peer review of "Distinct Changes in Gut Microbiota Are Associated with Estradiol-Mediated Protection from Diet-Induced Obesity in Female Mice"

_metabolites, 2021, doi:10.3390/metabo11080499_

Round 1
Reviewer 1 Report
This is an excellent manuscript reporting changes in gut microbiota populations induced by estradiol treatment. Data are convincing and absolutely new. The manuscript is well written and very clear.
Author Response
We thank Reviewer 1 for their comments. We have reviewed the manuscript and corrected minor spelling mistakes as suggested.
Reviewer 2 Report
Summary:
In the current manuscript, the authors focus on the role of estradiol/ovarian estrogens in obesity and metabolic changes and their possible associations with the gut microbiome. They have shown the ovariectomized adult mice with estradiol (E2) implants increase energy expenditure, physical activity, insulin sensitivity, and whole-body glucose, and decrease intestinal immune infiltrates and the tight junction protein occludin. On these results, the author postulates E2 exhibits a protective role in obesity, intestinal inflammation, and gut epithelial integrity. Further, they have shown altered microbiome profiles in response to E2 and HFD. Microbiome studies revealed changes and dysregulation of Coprobacillus, Lactococcus, and Akkermansia taxa and their correlations with metabolic dysregulation.
Comments,
- Check the name of the corresponding author!!
- In Figure 2B: Specify the time point on the image or in the figure legend
- In point 2-3, Line 127, the result is wrong or misinterpreted, all the parameters like food intake, physical activities, VO2 consumption, VCO2 production, and energy expenditure is increased but the heading indicated as reduced.
- Checking the serum protein expression of GDF15, which is a known inhibitor of food intake could be interesting to see in the standard and HFD mice after Estradiol implants.
- Line 148-149, misinterpretation of food intake at night after HFD feedings.
- Line 150-151, Both the result and citation are wrong.
- As the author explained in lines 154- 155, vehicle mice were less active during HFD, which does not make sense after looking at the y-axis (count/hr) from Figures 2B and 3B. it is also applicable to VCO2 production in E2 mice.
- Line 180-181, 185, 188-189, please provide the citation of the literature.
- Line 4B and C titles are misleading.
- Figure 4D please specify the source of the Estradiol detection.
- Figure 5 specifies the time point for insulin sensitivity.
- Figure 6A as DAPI stains all the nucleated cells including epithelial cells, it would be relevant to show H&E staining to better evaluate immuno-physiological changes. As the author shows only colonic regions, it would be interesting to see these changes in the small intestine. DAPI images look like more lymphoid aggregate or Peyer's patches than immune infiltrate. By adding the level of occludin is not sufficient to claim gut barrier permeability dysfunction, it would be interesting to assess the other tight junction proteins such as claudins, MLCK and further in-vivo FITC dextral gut permeability assay would strengthen the hypothesis.
- Figure 6D please show the immunohistology images of occludin and add data of ZO1.
- Line 234 about alpha diversity, the result is not clear as figure S4 A and B showed statistically significant differences in richness and evenness, so what is the intention of the author to aggregate the richness and evenness.
- Figure 8A, specify the diet and time-point of the experimental data. Overall figure 8 looks fussy and very hard to read. Please provide a clear image if possible or label it properly or manually. It would be clearer to show differentially abundance microbiome at phylum, order, class, family, and genus level separately and focus on relevant one.
- It would be interesting to perform a comparative prediction analysis of the functional metagenome (PICRUSt) of the gut bacterial microbiota to find metabolic pathways related with groups as compared in this analysis to study the effect of estradiol on obesity. Also, to check a few of the mRNA/protein expression of the genes which are associated with metabolic changes such as AMPK, PPARs, lipogenesis regulating genes, TCA cycle regulating genes. It could increase the impact of the study.
Round 2
Reviewer 2 Report
The author has incorporated the changes and given a possible explanation for most of the comments. So, I would like to accept this manuscript as it is after this revision.